# Effect of N on the Microstructure and Wear Resistance of 4Cr13 Corrosion-Resistant Plastic Mold Steel

**DOI:** 10.3390/ma17020308

**Published:** 2024-01-08

**Authors:** Yi Fan, Jian Zhou, Jinbo Gu, Hongxiao Chi, Dangshen Ma, Guanli Xie

**Affiliations:** Special Steel Department of Central Iron and Steel Research Institute (CISRI), Beijing 100081, China; fanyi98765@163.com (Y.F.); gujinboustb@163.com (J.G.); chihongxiao@163.com (H.C.); m2762@163.com (D.M.); xieguanli2021@163.com (G.X.)

**Keywords:** corrosion-resistant plastic mold steel, Cr_2_N, wear resistance, wear mechanism

## Abstract

The effect of N content on the microstructure and wear resistance of 4Cr13 corrosion-resistant plastic mold steel were investigated by scanning electron microscopy, transmission electron microscopy, X-ray diffraction, and tribometer. The results showed that the influence mechanism of nitrogen on the hardness of the test steels responded to the changes in the quenching temperature. When the quenching temperature was below 1050 °C, the solid solution strengthening of N played a dominant role as a wear mechanism, and as the N content increased, the hardness of the steel increased. When the quenching temperature was higher than 1050 °C, N increased the residual austenite content, resulting in a decrease in hardness. The addition of N reduced the optimal quenching temperature of the test steels. The N addition promoted the transformation of large-sized M_23_C_6_ to M_23_C_6_ and fine Cr_2_N, resulting in an increase in the hardness of the test steels. The influence on the wear resistance of the experimental steels differed according to the varied N contents. The addition of 0.1% N delayed the precipitation of large- sized particles in the second phase, increased the hardness of the steel, and reduced the degree of wear. However, an excessive addition of N (0.18%) led to the excessive precipitation of the second-phase particles, and the second-phase particles then gradually flaked during the wear process and continued to participate in the wear process as third-body abrasives, reducing wear resistance.

## 1. Introduction

Due to excellent hardenability, plasticity, toughness, and corrosion resistance, 4Cr13 corrosion-resistant plastic mold steel has been widely used in the production of high-precision injection molds, stamping molds, and cutting tools [1,2]. Melted plastic creates flow friction in molds, affecting the mold cavity during the pouring and injection processes. A certain amount of wear and tear on the mold cavity can also be caused by cured plastic, scratching it during the demolding process [3,4,5,6]. Therefore, as an important indicator, wear resistance has undeniable importance, affecting the service life of corrosion-resistant plastic mold steel. With the production and development of high-performance plastic products, as well as the working conditions of plastic molds becoming more complex and demanding [7,8], there is an urgent need to develop new plastic mold steels with higher wear resistance.

With the increasingly significant beneficial of nitrogen on the performance of steel, the addition of nitrogen to plastic mold steel has begun to receive attention and study [9,10,11]. The addition of nitrogen can partially replace expensive Ni and other alloy elements to improve the performance of mold steel [12,13].

At present, the research into the effects of N on Cr13-type corrosion-resistant plastic mold steel has been focused on the aspects of the microstructure, the mechanical properties, and corrosion resistance. Zhan Lichun [14] studied the effect of N content on the microstructure, the mechanical properties, and the corrosion resistance of 3Cr13 steel. The results showed that the strengthening effect of nitrogen on 3Cr13 steel occurred through solid solution strengthening and precipitation strengthening, which could significantly improve tempering hardness. In addition, N can refine the grain of the steel, which, according to the Hall–Petch relationship, plays a role through fine-grain strengthening. The contribution of nitrogen to tensile strength was higher than its contribution to yield strength. Wang Lichao [15] studied the effects of Mo and N on the microstructure and the properties of 3Cr14 steel. The results showed that N improved the hardness, impact toughness, tensile strength, and yield strength of steel through fine-grain strengthening and precipitation strengthening, while Mo and N could both increase the pitting-corrosion potential of 3Cr14 steel, which was beneficial to corrosion resistance. Xie Guanli [16] studied the effect of nitrogen content on the microstructure and the properties of 2Cr13 steel, based on 4Cr13 steel, by reducing carbon and increasing nitrogen. Reducing carbon and increasing nitrogen could reduce the microstructural segregation and the precipitation of M_23_C_6_ carbides and could also achieve excellent corrosion resistance and polishing performance. However, few studies about the effect of N on the wear resistance of 4Cr13 steel have been reported. Therefore, this study intended to examine the impact of N content on the microstructure and the wear resistance of 4Cr13 steel, as well as to clarify the wear mechanism.

## 2. Materials and Methods

The test material was 4Cr13 corrosion-resistant plastic mold steel that contained different mass fractions of N. The chemical composition is shown in Table 1, which is marked according to 0% N, 0.1% N, and 0.18% N test steel. The 4Cr13 corrosion-resistant plastic mold steel with different N contents was prepared using a 21 kg pressurized electroslag furnace. The ingot was annealed by slow cooling after stripping, and the Φ 16 mm round bar was forged after annealing. The forging temperature was 1140~1180 °C; the opening forging temperature was 1100~1150 °C; and the final forging temperature was ≥900 °C. After forging, slow-cooling annealing was adopted at 860 °C × 8 h. A quenching temperature of 920~1150 °C and a tempering temperature of 150~600 °C were selected as the heat-treatment-process temperatures.

A metallographic sample of Φ 16 × 10 mm was cut, ground, and polished on its surface using sandpaper; then, it was etched using 5 g CuCl_2_ + 30 mL HCl + 30 mL H_2_O + 25 mL alcohol solution. The microstructure and wear surface morphologies of the sample were detected using an inverted optical microscope (OM) (made of Olympus company, located in Tokyo, Japan) and a Quanta 650 thermal field emission-scanning-electron microscope (SEM) (made of FEI company, located in Hillsboro, OR, USA). The sample was processed into Φ 8 mm × 80 mm phase-analysis sample with a groove at one end, and the type and content of the second phase were quantitatively analyzed. Talos F200X transmission-electron-microscopy (TEM) (made of FEI company, located in Hillsboro, OR, USA) analysis was performed to determine the crystal structure and the types of precipitates. The wear experiment was conducted using the room-temperature reciprocating friction and wear module in the MFT-5000 multifunctional friction-and-wear testing machine(made of RTEC-Instruments, located in Silicon Valley, CA, USA). The grinding material was a Si_3_N_4_ ball, with a set load of 200 N, a frequency of 5 Hz, a duration of 60 min, and a stroke of 6 mm. Three-dimensional wear morphology was collected using a 10× white-light interferometer. The sample was ultrasonically cleaned and weighed before and after the abrasion test. Each test was repeated three times for each sample, and the average wear loss was calculated from the three sets of experimental data. The formulas of wear loss (Δm) and wear rate (δm) are shown below in Formulas (1) and (2), where $m_{1}$ and $m_{2}$ are the weights before and after wear [17].
(1)$${\Delta m=m}_{1}-m_{2}$$
(2)$$\delta m=\frac{m_{1}-m_{2}}{m_{1}}$$

The worn sample was measured for wear width, depth, and volume, using material surface wear measurement software and the unworn portion as the reference plane. Each wear track was measured three times, and the average value was taken. We used the TCFE10 database of Thermo-Calc software (version. 2023.1.111866-468) to calculate the precipitation-phase content of the test steels.

## 3. Results

### 3.1. Quenching Microstructure and Hardness

The microstructure of the test steels at different quenching temperatures is shown in Figure 1. As was observed, the quenching microstructure of the test steels was martensite, undissolved carbide, and residual austenite. When the quenching temperature was low, there were large, partially undissolved carbide particles in the steel from the annealing process. As the quenching temperature increased, the undissolved carbide particles became smaller and gradually dissolved into the matrix. At the same time, the martensitic microstructure became coarser with the increasing quenching temperature. The carbide size in the nitrogen-containing test steel was finer, and the dissolution rate was faster during quenching, so the dissolution tendency of carbides in the nitrogen-containing steel became more pronounced with the increasing temperature. Therefore, after quenching at 1050 °C, some carbides remained in the 0N steel, while the carbides in the 0.18 N steel were almost completely dissolved. The energy spectrum results (Figure 2) showed that the undissolved carbides in the test steel were predominantly Cr-rich carbides and were provisionally identified as M_23_C_6_ carbides.

Figure 3a shows the XRD patterns and the residual austenite content of the three test steels quenched at 1050 °C. After quenching, some of the austenite that had not been converted to martensite remained in the matrix, forming residual austenite. Nitrogen is an element that expands the austenite-phase region, lowers the Ms point of steel, and inhibits martensitic transformation more than carbon, thereby increasing the residual austenite content. Therefore, as the nitrogen content increased, the residual austenite content of the steel increased. The residual austenite content of 0.18 N steel is much higher than that of 0N steel when quenched at 1050 °C.

As shown in Figure 3b, increasing the quenching temperature enhanced the hardness of the test steel. This could be attributed to carbides and nitrides continuously melting into the matrix, causing a lattice distortion of the martensite and resulting in solution strengthening [18]. With the further increase in the quenching temperature, the hardness of the test steel showed a decreasing trend. On the one hand, the increase in the residual austenite after quenching reduced the hardness; on the other hand, the continuous dissolution of the carbides and the nitrides gradually weakened their pinning effect on the grain boundaries, promoting increases in grain size and resulting in decreased hardness [19].

When the temperature was below 1050 °C, the hardness of the nitrogen-containing steel was higher than that of the 0 N steel due to the dissolution of the carbides and the nitrides as well as the formation of interstitial solid solution strengthening. The nitrogen atoms in the nitrogen-containing steel solidly dissolved in the octahedral gap of FCC and underwent interstitial solid solution with the surrounding atoms, resulting in a non-cubic symmetry distortion and increasing the resistance to dislocation movement [20]. Meanwhile, the effect of the nitrogen atoms on the lattice asymmetry was stronger than that of the carbon atoms, and their strengthening effect on austenite was also stronger than that of the carbon atoms. Since addition of N resulted in a higher residual austenite content, the hardness of the nitrogen-containing test steel decreased rapidly, and the higher the N content, the more rapidly the hardness decreased when quenching above 1050 °C.

The three test steels reached their peak hardness values at 1050 °C, 1020 °C, and 1010 °C, respectively, among which the peak hardness of the 0 N steel was 60.1 HRC, that of the 0.1 N steel was 60.3 HRC, and that of the 0.18 N steel was 59.4 HRC. Based on the research results regarding the effect of the comprehensive quenching temperature on the quenched microstructure, hardness, and residual austenite content of the test steel, the quenching temperature that achieved the peak hardness of the test steel was selected as the final quenching temperature for the heat treatment process, namely, 1050 °C for the 0 N steel, 1020 °C for the 0.1 N steel, and 1010 °C for the 0.18 N steel.

### 3.2. Tempering Microstructure and Hardness

The tempering hardness and tempering microstructure of the test steels are shown in Figure 4. In Figure 4a, for tempering below 300 °C, the tempering hardness of the test steel gradually decreased with the increasing tempering temperature. At this time, the short-range diffusion of the carbon and nitrogen elements in the matrix had segregated, and the supersaturated carbon nitride had gradually dissolved, precipitating ε-carbonitride and forming tempered martensite. The carbon and nitrogen content in the matrix gradually decreased, resulting in a decrease in hardness. When the tempering temperature was between 300 °C and 500 °C, the tempering hardness gradually increased, reaching peak hardness at a temperature of 500 °C. In this temperature range, with the increase in tempering temperature, finely dispersed carbon nitrides precipitated, resulting in secondary hardening, while the ε-carbon nitride gradually transformed into alloy cementite [21]. Above 500 °C, the martensite decomposed; carbon nitrides aggregated and grew; and the microstructure gradually transformed into tempered sorbite, resulting in a sharp decrease in hardness. The secondary hardening tendency of the nitrogen-containing steels was significantly stronger than that of the non-nitrogen containing steels, primarily due to the fact that the co-precipitation of the nitrides and the carbon nitrides played a dispersive strengthening role in the nitrogen-containing steels during the tempering process, and the secondary hardening effect of nitrides was more pronounced [22]. The test steels reached the secondary hardening peaks around 500 °C, and the peak hardness values of the test steels were 0 N–56.8 HRC, 0.1 N–55 HRC, and 0.18 N–56.7 HRC.

The tempering microstructure of the test steel at 500 °C is shown in Figure 4(b1–b3). The tempering microstructure of the test steels consisted of martensite and the second phase. The second phase of the 0 N steel was predominantly composed of carbides, while the second phases of the 0.1 N and 0.18 N steels were predominantly composed of carbides or nitrides. There were large-particle carbides in the 0 N steel, and some carbides precipitated along the grain boundaries. The spherical carbides and nitrides in the nitrogen-containing steel were finer and more evenly distributed in the martensite matrix. With the increased nitrogen content, the number of carbides and nitrides also significantly increased.

The XRD diffraction patterns of the test steel tempering at 500 °C and the TEM morphology of the precipitates in 0.18 N steel are shown in Figure 5. After tempering at 500 °C, the main precipitate of the 0 N steel was M_23_C_6_, as well a small amount of M_3_C. The 0 N steel had first precipitated M_3_C during the tempering process. When the tempering temperature reached 500 °C, chromium replaced the iron in M_3_C, increasing its concentration and playing a dispersion-strengthening role [23]. The final microstructure contained M_23_C_6_ and a few M_3_C. The main precipitates of the nitrogen-containing steel were M_23_C_6_, Cr_2_N, and a small amount of M_3_C. The nitrogen-containing steel first formed fine hexagonal ε-[(Cr,Fe)_2_]N_1-x_ nitrides and ε-[(Cr,Fe)_2_]C carbides during the tempering process, and with the increase in the tempering temperature, the nitrides and the carbides transformed into cubic (Cr,Fe)_2_N and rhombohedral (Fe,Cr)_3_C [24]. Eventually, the precipitated phases in the nitrogen-containing steel contained primarily M_23_C_6_, Cr_2_N, and a few M_3_C.

As shown in Figure 5b, the precipitates in the nitrogen-containing steel were M_23_C_6_, Cr_2_N, and M_3_C. The size of the nitrogen-containing precipitate-phase Cr_2_N was around 200 nm, and that of the M_23_C_6_ precipitate phase was around 500 nm. The size of Cr_2_N was significantly smaller than that of the M_23_C_6_ precipitate phase and, as shown in Table 2, the higher the nitrogen content in the steel, the higher the proportion of Cr and N elements in the precipitate phase.

### 3.3. Wear Resistance

Figure 6 shows the change curve of the friction coefficient over time, as well as the wear loss and wear rate of the test steels. From the trend of the friction coefficient variation, it could be observed that the friction coefficient had underwent a transition from an initial unstable running-in stage to a stable growth stage. During the initial running-in stage, the friction coefficient changed significantly, and the sample surface had a certain roughness. The contact area between the grinding material and the sample gradually increased, resulting in an increase in the contact stress and the wear rate. With the rapid wear between the grinding material and the sample, the contact area became stable, the degree of wear gradually increased, and the friction coefficient steadily increased with the wear time [25]. The friction coefficient of the 0.1 N steel was significantly lower than that of the other two steels, and its wear resistance was the best. Compared with the 0 N steel, the friction coefficient of the 0.18 N steel fluctuated significantly; overall, the average friction coefficient of the 0.18 N steel was slightly lower than that of the 0 N steel.

As shown in Figure 6b, the wear loss and wear rate of the 0.1 N test steel were much lower than those of the other two test steels, and the wear resistance was the best. The 0.18 N steel had the highest nitrogen content, but its wear resistance was worse than that of the 0.1 N steel. It could be concluded that it may not necessarily be true that the higher the nitrogen content, the better the wear resistance. A moderate addition of nitrogen was beneficial for improving the wear resistance of the steel, but an excessive addition of nitrogen could reduce the wear resistance.

Figure 7 shows the morphology, the cross-sectional profile, the width, and the depth of the wear mark, as well as the wear volume. It could be observed that the test steel had shallow and deep wear marks on both sides, with the deepest roughly located at the center of the wear marks. The 0.1 N test steel had a more regular wear-mark profile, with a width, depth, and wear volume of 1.15 mm, 27 μm, and 0.040188 mm^3^. There were plastic deformation zones of bending and extrusion in the profiles of the 0 N and 0.18 N test steels. Whether it was the width, the depth, or the wear volume of the wear mark, the 0.1 N steel performed significantly better than the other two steels, corresponding to the wear-loss and wear-rate results.

Figure 8 shows the morphologies of the wear marks on the 0 N, 0.1 N, and 0.18 N test steels, as well as enlarged views of some areas. It could be observed that there were individual areas with severe wear, and normal wear areas on the wear surface. The test steels predominantly exhibited abrasive wear, with some adhesive and flaking wear. The 0 N steel showed the most severe wear, with a significant number of pits and debris in each area, accompanied by a large number of flaking layers. The wear marks of the 0.1 N steel were relatively shallow, with long, shallow micro-cutting marks and furrows in the normal-wear area along the sliding direction. The severe wear area had a few shallow flaking layers, presenting typical abrasive-wear characteristics. The surface wear morphology of the 0.18 N steel was primarily characterized by furrows, scaly flaking, and accumulated debris. There were large-scale flaking pits in the severe-wear area, as well as scaly flaking and more debris accumulation in the normal-wear area. From the state of the wear morphology, it could be observed that, due to the lack of strengthening and resistance of the carbides and the nitrides, the 0 N steel had the most severe wear, with a large number of flaking layers. The 0.18 N steel was next, with some flaking layers and wear debris caused by wear. The 0.1 N steel had the least degree of wear, with light, shallow wear marks and only a small amount of wear flaking layers remaining.

## 4. Discussion

Based on comprehensive data, such as the friction coefficient, wear loss, wear-mark width, and wear volume, it could be concluded that the wear resistance of both nitrogen-containing test steels was superior to that of the non-nitrogen-containing test steel as, generally speaking, the higher the hardness of a material, the better the wear resistance [26]. From this study, we found that the addition of different mass fractions of nitrogen resulted in an increase in the hardness of all the samples but did not equally improve the wear resistance. The wear resistance of the 0.18 N steel was better than that of the 0 N steel, but the wear resistance of the 0.18 N steel was lower than that of the 0.1 N steel. It could be speculated that the addition of different nitrogen quantities had led to different wear mechanisms. The influencing factors on wear mechanisms included hardness, structure composition, carbide, roughness, etc. [27,28,29,30,31]. Among them, the quantity, the morphology, and the distribution of carbides could have a significant impact on the wear resistance [32]. The presence of a second phase that was high in hardness, small in size, and had a dispersed distribution on its matrix could significantly improve the wear resistance of the material. In the early stages of wear, the carbides showed a resistance to dislocation movements in the matrix and interacted with dislocations to increase the critical shear stress, which strengthened the matrix and effectively resisted wear. During the wear process, the second phase in the matrix gradually exposed the surface of the matrix [33], and if the carbides were coarse or irregular in shape, it would cause stress concentration near the carbide, resulting in carbide detachment. Portions of the detached carbides remained in the friction contact area and continued to participate in wear as third-body abrasive particles, deteriorating the wear environment. At this time, the surface of the sample was not only interacting with the abrasive material but also contained the carbide debris that had remained on the surface [34]. Due to the lack of strengthening by the carbides and the nitrides, the hardness and the wear resistance of the 0 N steel were poor. Under external forces, the matrix was prone to deformation, and the deformation layer was relatively deep, while the large-sized, second phase of the steel consisted of intergranular precipitation. During the wear process, a large number of fatigue cracks had been generated, and within a short distance of diffusion, they formed a crack network, resulting in a large flaking area [35]. The detachment of the large carbide particles had participated in the wear process as a third-body abrasive and had gradually cut into the matrix, producing flaking layers. The flaking layers further intensified the abrasive wear, as abrasive particles, resulting in more severe wear. For the 0.1 N steel and 0.18 N steels, due to the addition of nitrogen, the type of the second phase formed in the steel had also changed accordingly. The molar-formation free energy of the second phase was calculated using the thermodynamic data of the second-phase element. The formula was as follows [36]:(3)$${∆G}_{298M_{x}X_{y}}={∆H}_{298}-298\left(∆S_{298}\right)=H_{298M_{x}X_{y}}-298(S_{298M_{x}X_{y}}-xS_{298M}-yS_{298X})$$
where *S*_298_ represents the standard molar entropy, *H*_298_ represents the standard molar enthalpy, and Δ*G*_298_ represents the standard molar-formation free energy. According to formula 3, the molar-formation free energy of Cr_2_N was −93.6 KJ, normalized to −46.8 KJ; that of Fe_3_C was 18.5 KJ, normalized to 6.2 KJ; and that of Cr_23_C_6_ was 405.7 KJ, normalized to 17.6 KJ. It could be assumed that Cr_2_N would be formed and precipitated preferentially during the experimental process in the nitrogen-added test steel.

Thermo-Calc software was used to calculate the equilibrium precipitation of the test steels, and the precipitation contents of Cr_2_N in the 0.1 N and 0.18 N steels were obtained as shown in Figure 9. It could be observed that when the nitrogen content was higher, more Cr_2_N precipitation was present. Due to the interfacial energy between the nitrogen compounds and the matrix being much smaller than that between carbides, the nitrogen compounds were more likely to form small strengthening phases [37]. The tiny dispersed nitrogen compounds were more advantageous for improving wear resistance. However, a higher nitrogen content could accelerate the diffusion of the nitrogen atoms, which would occur after obtaining relatively less energy, causing the nitrogen atoms to quickly migrate and react with the chromium atoms near the grain boundaries and shortening the nucleation time of the nitrides. At the same time, as the nitrogen atoms in the matrix further diffused towards the nucleation position of the nitrides, the growth of the nitrides accelerated showing that, overall, the higher the nitrogen content, the more obvious the precipitation and growth tendencies of the nitrides [38].

Figure 10 shows that during the wear process, under the action of a normal load and shear stress, the carbides and the nitrides gradually loosened and flaked, resulting in flaking pits. The flaking pits had then been continuously compressed and expanded, increasing the area size of the flaking pits.

Figure 11 shows the schematic diagram of the carbide flaking and flaking-pit expansion in the test steels. M_23_C_6_, with large sizes of about 300–500 nm, had a complex cubic structure with poor stability and a high susceptibility to growth. During the wear processes, large-sized M_23_C_6_ were more likely to loosen and detach from the matrix, forming flakes that were then continuously compressed and expanded, resulting in the expansion of the flaking area. The addition of the N element changed the precipitation-phase type from M_23_C_6_ to Cr_2_N and M_23_C_6_, in the steel. The size of Cr_2_N was much smaller than that of M_23_C_6_, and the small-sized Cr_2_N did not flake easily during the initial stages of wear, which enhanced its strengthening effect. In the later stages of wear, the pits produced by the flaking of small-sized Cr_2_N were relatively shallow, which manifested as shallower wear marks and lower wear volume.

There are a large number of fine, dispersed high-hardness carbides and nitrides in the 0.1 N and 0.18 N steels. The hardness values of these second phases were much higher than those of the matrix, which enhanced the critical stress required to bypass the dislocations of the second-phase particles and increased the hardness of the matrix. At this time, there were hard second-phases in the matrix. During the wear process, the matrix had high hardness and a strong ability to resist deformation, which could also provide stronger support for the second-phase particles to improve wear resistance. At the same time, research has shown that when there were micron-sized particles on the surface of the matrix and the particle height was higher than the surface of the matrix, the contact between the matrix and the particles could be reduced, preventing micro-cutting and protecting the matrix during wear. However, the interface between the matrix and the particles could be damaged during wear, resulting in particle detachment [39]. Therefore, in the early stages of wear, the second-phase particles in the nitrogen-containing steel resisted the micro-cutting of the friction pair, protecting the matrix and resulting in a low friction coefficient. Subsequently, the interface between the second-phase particles and the matrix became loose, and the second-phase particles gradually flaked and then joined the wear process as third-body abrasive particles, resulting in increased wear and higher friction coefficients.

With the increase in nitrogen content, the carbides and the nitrides aggregated and grew, and the precipitation quantity and size increased significantly. The large-sized second phases caused internal notches during the wear process, leading to concentrated stress and the initiation of source cracks, which could cause the matrix to break apart [40]. The excessive precipitation of the second phase led to an increase in the number of third-body abrasive particles that had separated from the matrix under normal stress during the later stages of wear. Therefore, a large amount of debris and more flaking layers were observed on the surface of the 0.18 N steel, and its degree of wear was more severe than that of the 0.1 N steel.

## 5. Conclusions

(1)The quenching microstructure of the test steel was martensite, carbide, and residual austenite. When the quenching temperature was below 1050 °C, due to the solid-solution-strengthening effect of N, the hardness of the nitrogen-containing steel was higher than that of the nitrogen-free steel. When the quenching temperature was higher than 1050 °C, the content of the residual austenite in the nitrogen-containing steel increased, and the hardness rapidly decreased, showing that the higher the N content, the more rapidly the hardness decreased.(2)With the addition of the N element, the type of second-phase material precipitated during tempering changed from M_23_C_6_ to M_23_C_6_ and Cr_2_N. The addition of the N element delayed the precipitation of large carbides, and the steel tended to precipitate fine Cr_2_N. The strengthening effect of the second phase resulted in greater hardness of the nitrogen-containing test steel.(3)The wear mechanism of the three test steels was predominantly abrasive wear, with some adhesive and flaking wear, and among the test steels, 0.1 N steel had the best wear resistance. The proper addition of N could not only increase the hardness of the steel but also delay the precipitation of the large-sized second phases, reducing the degree of wear. However, an excessive addition of N led to excessive precipitation of the second-phase particles, and the second-phase particles gradually flaked during the wear process before continuing to participate in the wear process, as third-body wear particles, resulting in internal notches and reducing wear resistance.

## Figures and Tables

**Figure 1 materials-17-00308-f001:**
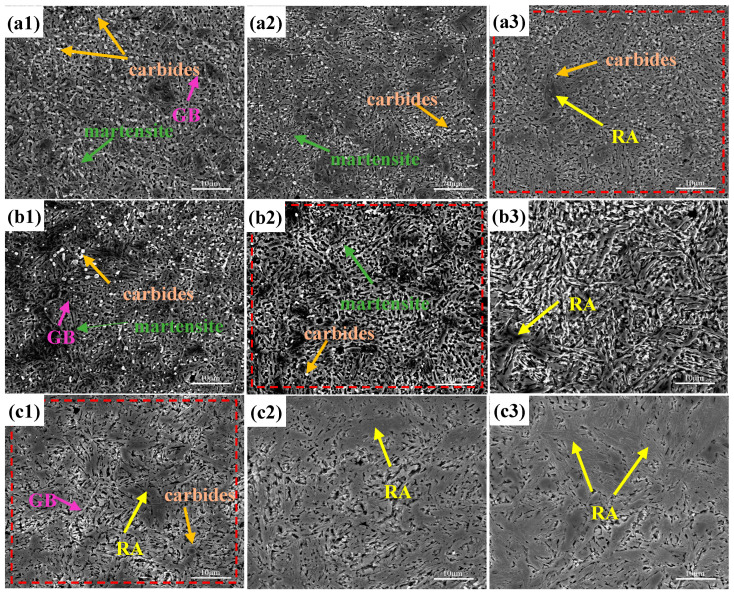
Quenching microstructure: (**a**) 0 N steel; (**b**) 0.1 N steel; (**c**) 0.18 N steel (GB—grain boundary, RA—residual austenite); **1**, **2**, and **3**, correspond, respectively, to quenching temperatures of 1000, 1020, and 1050 °C.

**Figure 2 materials-17-00308-f002:**
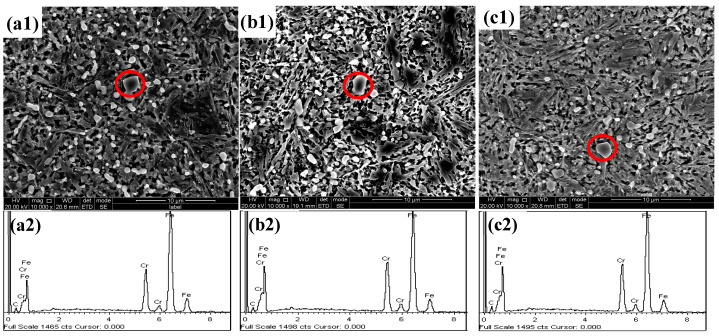
SEM of microstructure and EDS of undissolved carbide (**a1**,**a2**) 0 N steel—1050 °C; (**b1**,**b2**) 0.1 N steel—1020 °C; (**c1**,**c2**) 0.18 N steel—1000 °C.

**Figure 3 materials-17-00308-f003:**
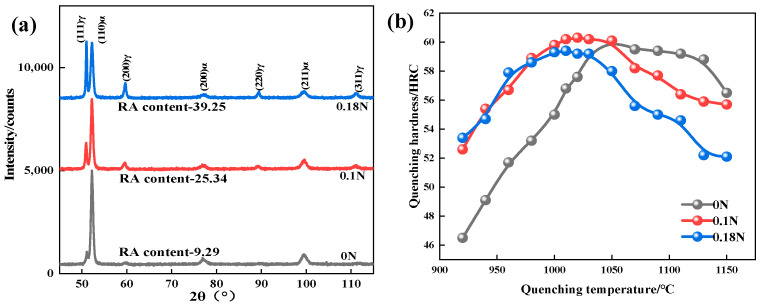
(**a**) XRD patterns after quenching at 1050 °C; (**b**) Quenching hardness curve.

**Figure 4 materials-17-00308-f004:**
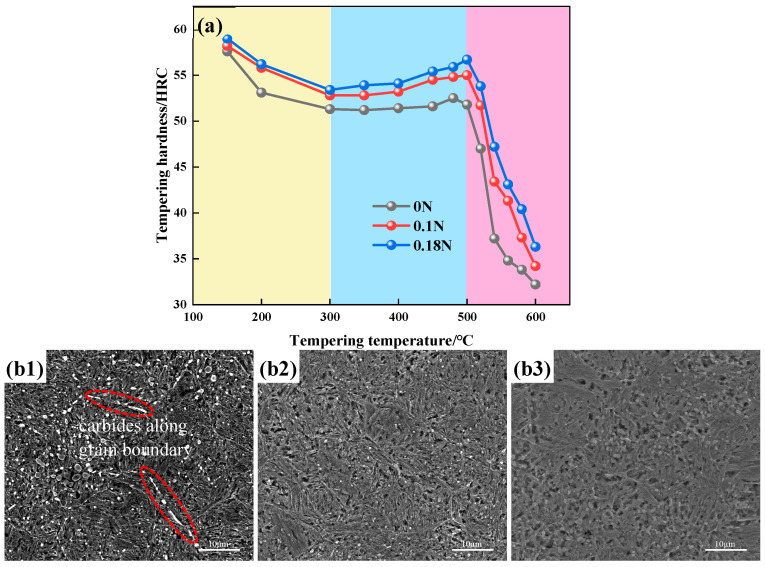
(**a**) Tempering hardness; (**b**) 500 °C tempering microstructure of test steels: **1**—0 N, **2**—0.1 N, **3**—0.18 N.

**Figure 5 materials-17-00308-f005:**
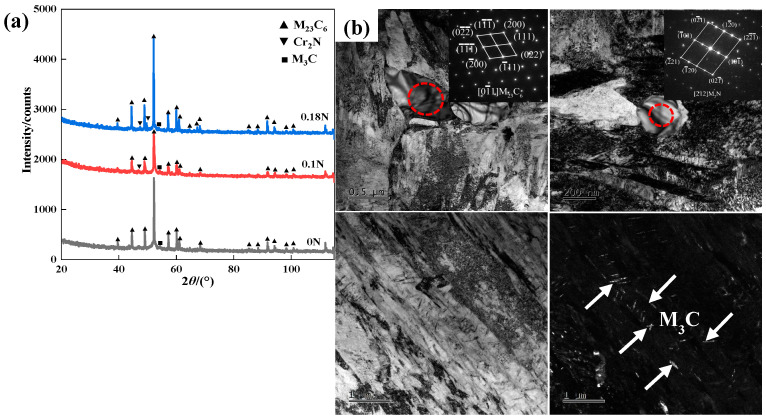
(**a**) XRD pattern after tempering at 500 °C; (**b**) TEM of precipitates in 0.18 N steel.

**Figure 6 materials-17-00308-f006:**
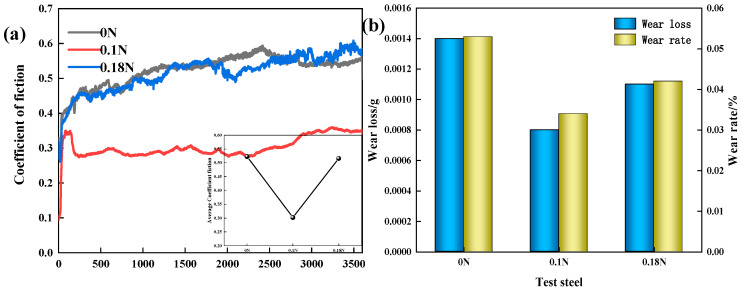
(**a**) Curve of friction coefficient; (**b**) Wear loss and wear rate.

**Figure 7 materials-17-00308-f007:**
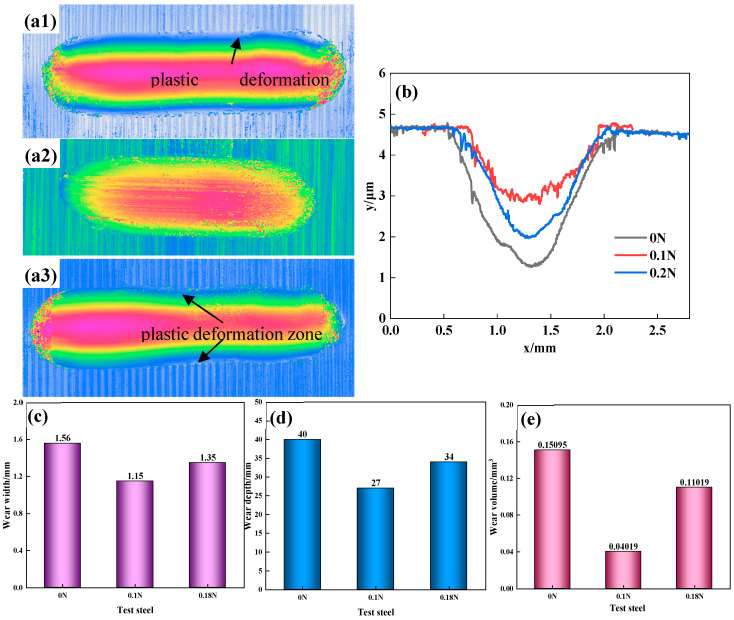
(**a**) Wear-mark morphology: **1**—0 N, **2**—0.1 N, **3**—0.18 N; (**b**) Wear-mark cross-section profile; (**c**) Wear width; (**d**) Wear depth; (**e**) wear volume.

**Figure 8 materials-17-00308-f008:**
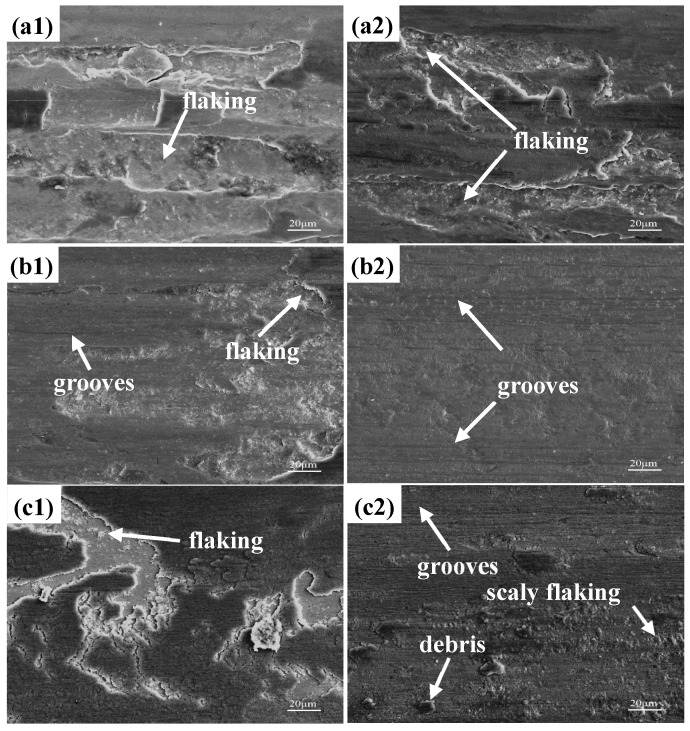
(**a1**,**a2**): Wear morphology of 0 N steel; (**b1**,**b2**): Wear morphology of 0.1 N steel; (**c1**,**c2**): Wear morphology of 0.18 N steel.

**Figure 9 materials-17-00308-f009:**
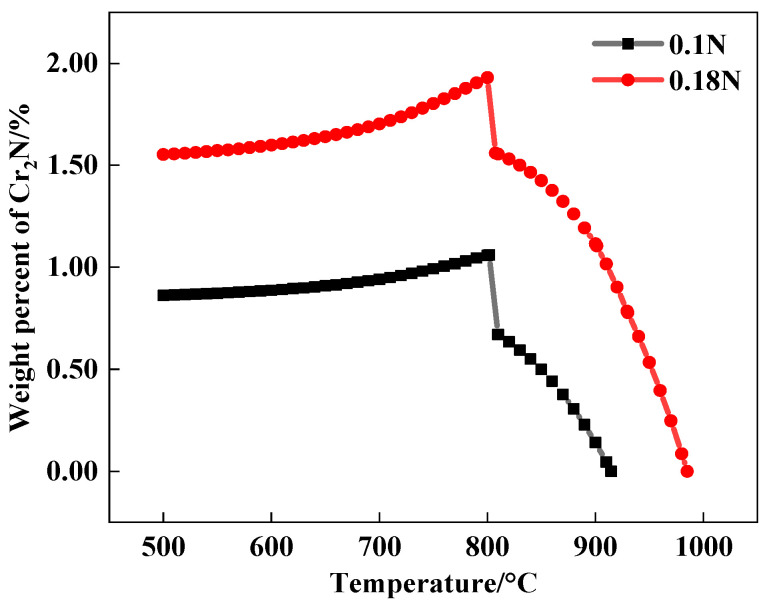
Precipitation content of Cr_2_N.

**Figure 10 materials-17-00308-f010:**
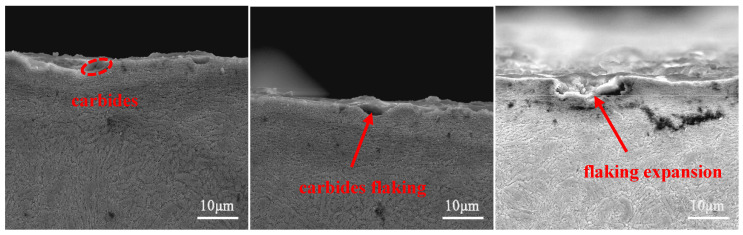
Process of generating flaking pits.

**Figure 11 materials-17-00308-f011:**
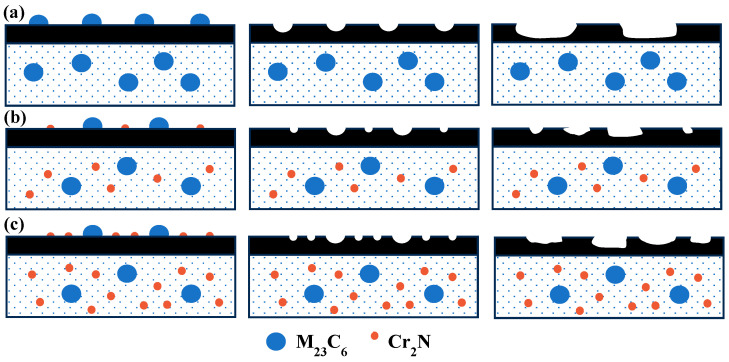
Schematic diagram of the formation of flaking pits: (**a**) 0 N steel; (**b**) 0.1 N steel; (**c**) 0.18 N steel.

**Table 1 materials-17-00308-t001:** Chemical composition of test steels (mass fraction, %).

Test Steel	C	Si	Mn	Cr	N
0 N	0.35	0.34	0.34	13.2	-
0.1 N	0.42	0.35	0.34	13.4	0.1
0.18 N	0.4	0.3	0.33	13.3	0.18

**Table 2 materials-17-00308-t002:** Mass fraction of precipitated-phase elements.

Test Steel	Mass Fraction of Precipitated-Phase Elements, wt%				
	Cr	Fe	Mn	N	∑
0 N	1.604	1.539	0.019	/	3.162
0.1 N	1.934	1.625	0.021	0.020	3.600
0.18 N	1.997	1.534	0.020	0.028	3.579

## Data Availability

Data are contained within the article.

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
