# Peer review of "Effect of N on the Microstructure and Wear Resistance of 4Cr13 Corrosion-Resistant Plastic Mold Steel"

_materials, 2024, doi:10.3390/ma17020308_

Round 1

Reviewer 1 Report

Comments and Suggestions for Authors

Dear Authors,

unfortunately, the quality of the English in your manuscript is not very good and therefore, difficult to read. The presented research is sound with regard to results, discussion and conclusions. However, the quality of the figures, in particular Fig. 1 and Fig. 9 could be improved. Also, i found another relevant literature (Yi Fan et al 2022 J. Phys.: Conf. Ser. 2342 012006), which is relevant to the design of the presented research and should be cited!

Regarding the experiments, I am not sure whether one wear experiment per sample status is sufficient.

I would suggest your research for publication after some small improvements.

With best regards.

Comments on the Quality of English Language

Has alrady been covered in the comments and suggestions to the Authors section and should be improved; e.g. proof reading by native speaker etc.!

Author Response

Thanks for your suggestions, the manuscript has been revised.

  • We have made efforts to improve the English quality of our manuscript.
  • We have changed Fig. 1 and Fig. 9 to improve the quality of the figures.
  • The literature (Yi Fan et al 2022 J. Phys.: Conf. Ser. 2342 012006), is created by myself, in order to enrich my research, I chose to cite more other articles.
  • Each wear experiment per sample status has been tested for three times. The data is stable and reliable.
  • According to PDF, we have carefully read and made revisions one by one.

Thank you and best regards.

Yours sincerely,

樊毅

通讯作者:

姓名:Yi Fan

电子邮件: fanyi98765@163.com

Reviewer 2 Report

Comments and Suggestions for Authors

The manuscript «The Effect of N on the Microstructure and Wear Resistance of 4Cr13 Corrosion-resistant Plastic Mold Steel» is well-organized and contains interesting results. I would recommend to accept it after minor revision. Here are the main comments.

1. Lines 31-33. Repeat of the sentence

2. Line 46. References on the papers on the positive effect of nickel on wear resistance of the steels are desired.

3. The introduction of the article could be enhanced by describing what type of microstructure and what hardness you are aiming for in order to obtain the highest possible wear resistance.

4. How N was introduced in the steel samples?

5. Line 114. What is the reason of complete dissolution of carbides in 0.18N steel? The concentration of carbides and nitrides should be higher than in 0N steel and thus, their complete dissolution should be more complicated?

6. It seems that Figure 3 is not very informative. Why the chemical composition of the strengthening carbides or nitrides is important here?

7. XRD is not described in Materials and Methods subsection

8. Figure 4b. According to the graph, different quenching temperatures were chosen for experiment. Some discussion on quenching temperature selection is desired in Materials and Methods subsection.

9. Figure 6 b. How the dark field image was taken?

10. Please describe the methodology of wear rate calculation.

11. Line 250. Reference to the Figure 9 is needed here.

Reviewer 3 Report

Comments and Suggestions for Authors

The article materials-2769726-peer-review-v1. The Effect of N on the Microstructure and Wear Resistance of 2 4Cr13 Corrosion-resistant Plastic Mold Steel. The author has correlated the aspect of hardness in improving the wear properties of steel. However, the ability of a steel to work harden is important in enhancing the wear resistance, because it is the surface hardness that determines the interaction between the abrasive and the steel. The manuscript is overall well written but need some minor correction for final recommendation for publication. Please see enclosed my report.

Comments on the Quality of English Language

Moderate editing of the English language required

Reviewer 4 Report

Comments and Suggestions for Authors

The paper is very interesting, but there are several points to be clarified:

11)    The quenching and tempering process are not described in section 2.

22)    The Thermo-Cal calculations are not described in section 2. What database?

33)    Lines 81-83, text is not clear. For example, a microstructure is observed and not detected.

44)    Figure 1 is not relevant.

55)    The authors reported a lath martensite, but it seems martensite plates in Fig. 2.

66)    The authors should read a paper of this journal and the author instructions.

77)    A printed point is missing in all figure captions.

88)    The correct term is XRD patterns.

99)    Why M23C6 are not detected in XRD patterns of Fig. 4 (a).

110) English need revision in all the manuscript.

111) Lines 312-320, the Cr2N precipitation increased with the increase in N content of teel because of the higher supersaturation and thus, higher driving force.

112) Conclusion 2 is not clear “changes from M23C6 to M23C6+Cr2N”. It seems that Cr2N precipitates.

113) Alpha prime is used to designate the martensite, avoiding to confuse with alfa ferrite.

114) Paper shoul be revise carefully. There are many errors.

Comments on the Quality of English Language

The paper is very interesting, but there are several points to be clarified:

1)    The quenching and tempering process are not described in section 2.

2)    The Thermo-Cal calculations are not described in section 2. What database?

3)    Lines 81-83, text is not clear. For example, a microstructure is observed and not detected.

4)    Figure 1 is not relevant.

5)    The authors reported a lath martensite, but it seems martensite plates in Fig. 2.

6)    The authors should read a paper of this journal and the author instructions.

7)    A printed point is missing in all figure captions.

8)    The correct term is XRD patterns.

9)    Why M23C6 are not detected in XRD patterns of Fig. 4 (a).

10) English need revision in all the manuscript.

11) Lines 312-320, the Cr2N precipitation increased with the increase in N content of teel because of the higher supersaturation and thus, higher driving force.

12) Conclusion 2 is not clear “changes from M23C6 to M23C6+Cr2N”. It seems that Cr2N precipitates.

13) Alpha prime is used to designate the martensite, avoiding to confuse with alfa ferrite.

14) Paper shoul be revise carefully. There are many errors.

Round 2

Reviewer 1 Report

Comments and Suggestions for Authors

Thanks for editing the manuscript, I will suggest it for publication in its present form.

Reviewer 3 Report

Comments and Suggestions for Authors

Dear Authors, Thank you for the revision. I recommend paper for publication

Comments on the Quality of English Language

Minor editing shall be taken care of for English 

Reviewer 4 Report

Comments and Suggestions for Authors

Paper was improved to be accepted.